# To Be a Champion of the 24-h Ultramarathon Race. If Not the Heart ... Mosaic Theory?

**DOI:** 10.3390/ijerph18052371

**Published:** 2021-03-01

**Authors:** Robert Gajda, Aleksandra Samełko, Miłosz Czuba, Agnieszka Piotrowska-Nowak, Katarzyna Tońska, Cezary Żekanowski, Anna Klisiewicz, Wojciech Drygas, Anita Gębska-Kuczerowska, Jacek Gajda, Beat Knechtle, Jakub Grzegorz Adamczyk

**Affiliations:** 1Center for Sports Cardiology, Gajda-Med Medical Center in Pułtusk, 06-100 Pułtusk, Poland; j.gajda@gajdamed.pl; 2Department of Pedagogy and Psychology of Physical Culture, Faculty of Physical Education, Józef Piłsudski University of Physical Education in Warsaw, Marymoncka St. 34, 00-968 Warsaw, Poland; al.samelko@gmail.com; 3Department of Applied and Clinical Physiology, Collegium Medicum University of Zielona Gora, 28 Zyty St., 65-417 Zielona Gora, Poland; czubamilosz@gmail.com; 4Department of Kinesiology, Institute of Sport, 2 Trylogii St., 01-982 Warsaw, Poland; 5Institute of Genetics and Biotechnology, Faculty of Biology, University of Warsaw, Pawinskiego 5a Street, 02-106 Warsaw, Poland; apiotrowska@biol.uw.edu.pl (A.P.-N.); kaska@igib.uw.edu.pl (K.T.); 6Laboratory of Neurogenetics, Mossakowski Medical Research Institute, Polish Academy of Sciences, ul. Pawinskiego 5, 02-106 Warszawa, Poland; c.zekanowski@imdik.pan.pl; 7The Cardinal Stefan Wyszyński National Institute of Cardiology, ul. Alpejska 42, 04-628 Warszawa, Poland; aklisiewicz@ikard.pl (A.K.); wdrygas@ikard.pl (W.D.); 8Department of Preventive Medicine, Faculty of Health, Medical University of Lodz, ul. Lucjana Żeligowskiego 7/9, 90-752 Łódź, Poland; 9Faculty of Medicine, Collegium Medicum, Cardinal Stefan Wyszyński University, Kazimierza Wóycickiego 1/3, 01-938 Warsaw, Poland; agkucz@vp.pl; 10Institute of Primary Care, University of Zurich, 8091 Zurich, Switzerland; beat.knechtle@hispeed.ch; 11Medbase St. Gallen Am Vadianplatz, 9000 St. Gallen, Switzerland; 12Department of Theory of Sport, Faculty of Physical Education, Józef Piłsudski University of Physical Education in Warsaw, Marymoncka St. 34, 00-968 Warsaw, Poland; jakub.adamczyk@awf.edu.pl

**Keywords:** ultramarathon, mental and physical stress markers, VO_2_max, polymorphism, psychological features, pain resistance, body composition, cardiopulmonary exercise test (CPET), team support, winner’s gen

## Abstract

This comprehensive case analysis aimed to identify the features enabling a runner to achieve championship in 24-h ultramarathon (UM) races. A 36-year-old, multiple medalist of the World Championships in 24-h running, was assessed before, one and 10 days after a 24-h run. Results of his extensive laboratory and cardiological diagnostics with transthoracic echocardiography (TTE) and a one-time cardiopulmonary exercise test (CPET) were analyzed. After 12 h of running (approximately 130 km), the athlete experienced an increasing pain in the right knee. His baseline clinical data were within the normal range. High physical efficiency in CPET (VO_2_max 63 mL/kg/min) was similar to the average achieved by other ultramarathoners who had significantly worse results. Thus, we also performed genetic tests and assessed his psychological profile, body composition, and markers of physical and mental stress (serotonin, cortisol, epinephrine, prolactin, testosterone, and luteinizing hormone). The athlete had a mtDNA haplogroup H (HV0a1 subgroup, belonging to the HV cluster), characteristic of athletes with the highest endurance. Psychological studies have shown high and very high intensity of the properties of individual scales of the tools used mental resilience (62–100% depending on the scale), openness to experience (10th sten), coherence (10th sten), positive perfectionism (100%) and overall hope for success score (10th sten). The athlete himself considers the commitment and mental support of his team to be a significant factor of his success. Body composition assessment (%fat 13.9) and the level of stress markers were unremarkable. The tested athlete showed a number of features of the champions of ultramarathon runs, such as: inborn predispositions, mental traits, level of training, and resistance to pain. However, none of these features are reserved exclusively for “champions”. Team support’s participation cannot be underestimated. The factors that guarantee the success of this elite 24-h UM runner go far beyond physiological and psychological explanations. Further studies are needed to identify individual elements of the putative “mosaic theory of being a champion”.

## 1. Introduction

Attempts to find the key to success in each sports discipline date from the beginning of sports competitions [1]. The key to achieving championship results in endurance sports, especially in ultramarathon (UM) runs, seems to be particularly difficult to identify due to the multifactorial nature of a success [2]. In 24-h and longer running races, in addition to the normal running endurance, the competitors must be resistant to lack of sleep and pain, have special endurance after years of preparation, and maintain a generally good health with resistance to changing weather conditions, infections, and many other factors, as well as support, special commitment, and professionalism of the team. The professionalism of the team should be understood as several behaviors that contribute to the creation of optimal conditions for the runner during the marathon (appropriate temperature and volume of meals served, strict supervision of the runner’s service time, ability to motivate, and so on) [3].

According to Jason Koop, a coaching director and the author of “Training Essentials for Ultrarunning”, despite their diverse backgrounds, elite ultrarunners share the following three common characteristics: (1) talent: all elite ultrarunners (and elite endurance athletes in general) have high genetic potential for aerobic power; (2) toughness: which can be harnessed with focused training; and (3) emotional engagement: the best runners have an emotional attachment to the races they compete in. They perform best when they have a genuine, visceral attraction to a race, the type of attraction that makes them excited, giddy, and apprehensive simultaneously. The best UM runners possess the highest levels of talent, toughness, and emotional engagement. For the rest, the key is to optimize what they have by tapping into and leveraging and developing their own innate talent, toughness, and emotional engagement [4]. In the case of endurance sports, three physiological variables such as maximum oxygen consumption (VO_2max_), workload at lactate (LA) threshold (LT), and efficiency (the amount of oxygen needed to generate a certain running speed) play key roles [5]. However, while the athletes run with approximately 90% VO_2max_ in a 10-km run [1], this intensity decreases to approximately 75–85% VO_2max_ as well as their %VO_2max_ at 42 km (i.e., the marathon distance) [5]. The body’s adaptations to training that contribute to high VO_2max_ values include an increased cardiac stroke volume, an increased blood volume, and an increased capillary and mitochondrial density in the trained muscles [6]. In runs shorter than an UM, the most dominant factor is a high stroke volume [7]. Apart from sports training, which has a fundamental impact on the parameters mentioned, genetic predispositions, which until now have not been unequivocally defined as “champion’s genes,” should also be of importance [8].

A fundamental question is the role of genetics in the attainment of world class status and truly elite athletic performance [9]. Several studies have reported that the key elements of the response to training in sedentary persons are widely variable and have a genetic component [8]. Genetic potential plays a role, and mtDNA variability, influencing mitochondrial activity, is expected to be of importance. Hence, interest in mtDNA seems justified. Assuming that there is a limited set of genes causally related to sports performance, the interdependence of the genetic regulatory networks of the cell and the interactions at higher levels of the organization from cell to organism mean that a large part of heritability may result from genome-wide variability [10]. However, to date, there are no genetic markers identified in humans that have been clearly shown to be more frequent in elite endurance athletes [11].

However, the usual psychological, sociological, economic, and cultural factors may be much more important than the “genes’” themselves in determining the factors to achieve a championship in UM running [12]. For example, Burns et al. investigated Olympic and Paralympic champions who attributed their success to psychological factors [13]. Some models try to explain how environmental conditions, sleep deprivation/mental fatigue, pain-killers or psychostimulants, cognitive or nutritional strategies may affect UM performance [14].

In the ability to run up to 1000 km in one stage, fatigue resistance is critical. Fatigue is generally defined as strength loss, which is known to be dependent on the type of exercise [14]. The evaluation of neuromuscular function requires measurements of maximal voluntary contractions and maximal electrical/magnetic stimulations to elucidate the factors affecting the central nervous system and the muscles implicated by fatigue. However, such measurements do not necessarily predict how muscle function may influence ultra-endurance running and whether this has an effect on speed regulation during a real competition. The nature of the relationship between fatigue as measured using maximal contractions/stimulation and submaximal performance limitation/regulation is questionable. Millet et al. suggested a holistic flush model that can be applied to all endurance activities, but this is specifically adapted to ultra-endurance running [14]. This model had the following four components: the ball-cock, which can be compared with the rate of perceived exertion and the increase or decrease based on it; the filling rate; the water evacuated through the waste pipe; and a security reserve that allows the subject to prevent physiological damage. They suggested that central regulation is not only based on afferent signals arising from the muscles and peripheral organs but also dependent on peripheral fatigue and spinal/supraspinal inhibition (or disfacilitation) since these alterations imply a higher central drive for a given power output [14]. This holistic model further explained how environmental conditions, sleep deprivation/mental fatigue, painkillers or psychostimulants, and cognitive or nutritional strategies may affect ultra-running performance. In conclusion, the following two phenomena categorize fatigue: the diminution of muscular force (the physical aspect that can be measured and compared) and the sensation of fatigue (a psychic aspect that could not be measured).

The performance of elite athletes is likely to defy the types of easy explanations sought by scientific reductionism and remain an important puzzle for those interested in physiological integration well into the future [9]. Therefore, further research is needed to determine the factors that allow UM runners to achieve championship in a 24-h run.

In our previous papers, we analyzed several factors related to physiological and biochemical adaptation to extreme endurance exercise in order to demonstrate that organism of elite, endurance-trained athletes is able to adapt to strenuous long-lasting run (marathon, UM, triathlon) or swimming [15,16]. In some of our previous publications, we used the unique occasion to analyze various aspects of adaptation to 24-h UM in an elite runner, a previous winner of several prestigious international UM during the National Championship in 24-h run [17,18]. The aim of the present study was to analyze several key factors responsible for champion achievements in a 24-h UM race in the same very successful athlete. The analyses comprise body measurements, cardiac morphology and function, cardiopulmonary exercise test, markers of physical and mental stress, genetic factors, and battery of psychological tests. We also included a live story interview and analyzed the role of team support. We were not able to find in the available scientific literature a similar comprehensive study describing so many aspects of potential UM performance in a champion ultra-endurance runner.

Although we acknowledge that a case study does not allow us to generalize the results obtained, we believe that a thorough investigation of one, but outstanding, athlete makes an important step in the further discussion about ultra-endurance performance, bringing us closer to the relevant answer of the question “What are the key factors of elite performance in 24-h UM run?”.

## 2. Materials and Methods

### 2.1. Sports Biography and Main Achievements

In this case study, we evaluated a multi-awarded Polish UM runner (36 years, body height, 1.73 m; body mass, 63 kg; body mass index, 21.05 kg/m^2^) on the day of a competition. He has a sedentary profession and worked Monday to Friday from 8:00 to 16:00 h and runs regularly for 20 years and approximately 100,000 km in his lifetime. As part of his daily training over the past year, he had run an average of 22 km/day from Monday to Friday and approximately 37 km on Saturdays and Sundays. His activities included swimming, gymnastics, and cross-training in the gym. For many years, he represented Poland in UM races, and his performance over the past several years can be found on his official website [19]. This athlete participated in approximately 50 UMs, the longest of which was a 48-h run (362-km crown distance, 24-h UM). He had never been injured or seriously ill. He was one of the most successful UM runners in the world, a two-time Polish champion in the 24-h UM (2017, he ran 258.228 km during the UM), the winner of the Spartathlon in Greece (2016, 246 km), and the runner who set the Polish record for 12-h races in 2014 (145.572 km). He won medals in four 24-h UM World Championships. In 2019, he won the 48-h race in Athens (362 km). Other best personal results: 100 km, 7 min 37 s; marathon: 2 min 42 s; half marathon: 1 min 16 s; 10 km: 35 min 28 s. The top ten UM races are listed in Table 1.

At 2 weeks before the start of the 24-h UM, the athlete gradually reduced his training activities until the competition. The competition referred to in this study started at 12:00 h on 8 April 2017. All participants ran on a 2-km loop (exact distance 1.984 km) in a clockwise direction. On one of the straight sections was a tent where an athlete could stop for a meal, a short rest, or basic hygiene. After 12 h, our subject stopped for 12–15 min for a warm meal and to rehydrate, changed his shoes, and used the toilet three times (2–3 min each). The primary goal of the study was the cardiovascular assessment (separate article) [17]. For the first time at the finish line, our subject experienced pain in the right knee starting after approximately 12 h of running (approximately 130 km) and intensified up to the end, thus affecting his performance. Magnetic resonance imaging on the right knee one day after the run showed a general overload and degenerative as well as specific features associated with “turning to the right” on the run loop (separate article) [18]. These competitions comprised the official Polish championships of the 24-h UM.

### 2.2. Study Protocol

Transthoracic echocardiography (TTE) and blood tests were performed three times and other tests once (Table 2).

#### 2.2.1. Laboratory Examinations

Blood samples were collected from the median cubital vein with a Kima closed blood collection system. Samples were collected into the following tubes: trisodium potassium edetate (EDTA K3) tubes for blood morphology testing, 3.2% sodium citrate tubes for coagulation tests, and clotting activator tubes for biochemical and immunochemical tests. Samples for biochemical and coagulation tests were spun in a Nuve NF 800 (Henderson Biomedical, Ankara, Turkey) centrifuge for 10 min at 3000 rpm. Blood morphology was analyzed using a SYSMEX XS 1000i analyzer (Sysmex Corporation, Kobe, Japan). Biochemical tests were conducted using serum (obtained by centrifugation) and a Roche Integra 400 PLUS (Roche Diagnostics, Basel, Switzerland). Immunochemical tests were performed using a Roche Cobas E 411 (Roche Diagnostics, Basel, Switzerland). Coagulation tests were conducted using plasma and performed using a Bioksel 6000 (Bio-Ksel, Grudziądz, Poland). The parameters determined were subjected to internal and external laboratory controls (COBJ in DL; EQuas of Bio-Rad, Hercules, CA, USA).

#### 2.2.2. Cardiopulmonary Exercise Test (CPET) and Body Composition Assessment

Two weeks before the run, the participant performed a CPET to assess his maximal oxygen uptake (VO2max) and LT. Before breakfast and after an overnight fast, body measurements were taken. Body height was measured to the nearest 0.1 cm using digital stadiometer BSM 170 (Inbody Co., Seoul, Korea) and body mass was determined to the nearest 0.1 kg using a medical scale (InBody 570, InBody Co., Seoul, Korea). Body composition was evaluated using dual-energy X-ray absorptiometry (DEXA) (Discovery Wi, HOLOGIC, Bedford, MA, USA). Two hours after a light, mixed breakfast, the CPET protocol was administered. As ultra-endurance runners mostly train with high volume training at low to moderate intensities, without speed training targeted at the development of motor coordination required for fast running, the test consisted of a combination of treadmill speed and inclination. The CPET (Pulsar, HP Cosmos, Nussdorf-Traunstein, Germany) started with a speed of 6 km/h and 0° inclination that was increased by 2 km/h every 3 min until the speed reached 14 km/h. Then, keeping the speed steady at 14 km/h, the inclination was increased by 2.5% every 3 min and the run continued to volitional exhaustion.

Given that the inclination on the treadmill was increased during the CPET test, and to make the interpretation of the result easier to understand for coaches, the intensity was expressed as velocity (v) with grade (g), and power (WR). Power was calculated using the MetaSoft software (Cortex, Leipzig, Germany) with the following formula: WR = (1.065 + 0.0511 * g + 9.322 * 10 − 4 * g2) * v * w/4, where WR is the workload in W, g is grade in %, v is velocity in km/h, and w is weight in kg.

During the test, heart rate, minute ventilation (VE), oxygen uptake (VO_2_), and expired carbon dioxide (CO_2_) were measured continuously using the MetaLyzer 3B-2R spiroergometer (Cortex, Germany) in the breath-by-breath mode. The criteria of reaching VO_2_max included (1) a plateau in VO_2_ at rising WR (ΔVO_2_ ≤ 150 mL/min at VO_2_peak), (2) maximal respiratory exchange ratio ≥ 1.1, and (3) LA concentration ≥ 8 mmol/L.

Fingertip capillary blood samples for the assessment of LA concentration (Biosen C-line Clinic, EKF-diagnostic GmbH, Barleben, Germany) were drawn at rest, at the end of each step of the test, and at the first 3 min of recovery. LT was determined by the D-max method [20]. VO_2_max was determined based on the following criteria: (1) a plateau in VO_2_ at rising workload (ΔVO_2_ ≤ 150 mL/min at VO_2_peak), (2) maximal respiratory exchange ratio ≥ 1.1, and (3) LA concentration ≥ 8 mmol/L. All gas exchange data were time-averaged using 15-s intervals to examine the VO_2_ plateau. Additionally, capillary test and post-exercise blood samples were used to determine acid-base equilibrium (RapidLab 248, Bayer Diagnostics, Leverkusen, Germany).

#### 2.2.3. Transthoracic Echocardiography

Our subject underwent three complete TTE examinations (3 days before, 1 day after, and 10 days after the race) using a GE Medical System Vivid 7 (Chicago, IL, USA) with a 2.5-MHz transducer by the same investigator. M-mode, two-dimensional imaging, and Doppler techniques were performed. Left ventricular (LV) end-systolic and LV end-diastolic volumes, interventricular septal diastolic diameter, and posterior wall thickness diameter were measured. LV systolic function was evaluated by LV ejection fraction (LVEF) and longitudinal strain [global longitudinal strain (GLS)]. LV diastolic function was evaluated using mitral inflow velocities and tissue Doppler imaging (TDI) values. The right ventricular end-diastolic diameter from the parasternal long-axis view and the tricuspid lateral annular systolic velocity wave (S’RV) were measured using TDI. The right atrial area (RAA) and left atrial volume index were calculated using the body surface area.

#### 2.2.4. Genetic Examination

DNA was isolated from peripheral blood using a standard salting out method [21]. The mitochondrial genome sequence and variant heteroplasmy were analyzed using high-throughput sequencing as described previously [22,23]. In brief, whole mtDNA was amplified from total DNA using a single primer pair and long-range polymerase chain reaction (PCR). PCR product was subsequently purified and quantified before proceeding to DNA library preparation for next generation sequencing. Indexed mtDNA library was prepared for sequencing on a MiSeq instrument with a 251-cycle paired-end read chemistry using Illumina’s Nextera XT protocol. The raw data were initially automatically processed on platform before further bioinformatic analysis of FASTQ files using CLC genomics workbench software (CLC bio, Qiagen. Aarhus, Denmark). The workflow consisted of quality control of sequencing reads, mapping to the human mtDNA reference sequence (rCRS, GenBank sequence NC_012920), variant detection, annotation and evaluation based on the strategy described in detail previously [22]. The mitochondrial haplogroup assignment was performed using a full set of variants identified in mtDNA of the subject and the Mitomaster tool [24]. The variant population frequency was obtained using a current human mtDNA sequence dataset deposited in GenBank and available from MITOMAP database [24]. Bioinformatic analysis of next generation sequencing (NGS) data in search of large mtDNA deletions was performed using CLC genomics workbench InDels and structural variants and eKLIPse tools [25].

#### 2.2.5. Psychological Tools

In the study of the psychological profile, we used semi-structured interviews, standardized psychological questionnaires, and experimental versions of tools with particular emphasis on athletes. In the psychological part, research tools were used, some of which were standardized questionnaires adapted to Polish conditions together with population norms. Two questionnaires were experimental versions used with particular emphasis on athletes (without sten norms) and interview as a qualitative research tool.

Personality inventory (NEO-FFI) is a personality questionnaire based on the five-factor model of traits: neuroticism, extraversion, openness to experience, agreeableness, and conscientiousness [26]. In the Polish adaptation, the reliability of the scales and accuracy are sufficient for scientific research [27]. The Sense of Coherence Questionnaire (SOC 29) is the Polish adaptation of the tool by Antonovsky [28]. It defines the sense of coherence using three components: sense of comprehensibility, manageability, and meaningfulness. The reliability of the questionnaire is very good, and its accuracy has been demonstrated [29]. The Hope for Success Questionnaire (KNS) measures the strength of expectations of the positive effects of one’s own actions. It is a Polish adaptation of Snyder et al. [30]. The tool contains two constructs: agency (willpower) and pathway (ability to find solutions). The psychometric properties are sufficient to conduct research [31]. The Generalized Self-Efficiency Scale (GSES) measures the strength of an individual’s belief in the effectiveness of coping with difficult situations. The authors of the Polish version of Schwarzer, Jerusalem, Juczyński [32] considered the accuracy and reliability of the questionnaire to be satisfactory. Scheler Value Scale (SVS) is used to evaluate the hierarchy and importance of values for an individual. A tool adapted from Brzozowski [33] analyzes four groups of fundamental values: hedonistic, vital, spiritual, and sacred, of which spiritual values are divided into aesthetic, truth, and moral values [34]. In research, it is possible to undertake interpretation of the values of the factor subscales, secular, religious, fitness, and physical strength and endurance. The tested accuracy and reliability turned out to be satisfactory [33].

Life Orientation Test (LOT-R) is an indicator of optimism [35] and the adaptive version [36] regards optimism as a dispositional trait expressed through the expectation of positive events; this tool tested for reliability and validity. In the study, we used Polish adaptation of the questionnaire of Sport Mental Toughness (SMTQ) [37]. The questionnaire includes the following scales: self-confidence, effectiveness, emotional control, and completing tasks. The presented dependencies seem to confirm, at least partially, the reliability and topicality of the Polish version *SMTQ* [38]. The Perfectionism in Sport Questionnaire by Stolarski and Waleriańczyk [39] is an authorial-experimental version, containing two scales: positive and negative perfectionisms. Psychologists have proven the two-factoriality of the variable model, which means there is no significant relationship between the result achieved in positive and negative perfectionisms [40]. The Live Story Interview by Dan McAdams [41] is narrative, in which the research material denotes the story told by the subject and the meaning he gives it. It can be standardized and is repeatable. In the Polish version, a cultural adaptation and stylistic correction of the text were made. For the purposes of this study, an interview on the history of life in the context of development of a sports career was used. The recorded material (with the consent of the respondent) was divided into the suggested significance indicators’ communion and agency. Communion means a feeling of unity with others, and agency proves expansion and divisions, including separating oneself from others [42].

#### 2.2.6. Ethical Approval

This study was approved by the ethical review board of the bioethics committee of the Healthy Lifestyle Foundation in Pułtusk (EC 3/2017/medicine/sports, approval date: 30 March 2017). All experiments and procedures were conducted in accordance with the Declaration of Helsinki. Our subject provided his written informed consent prior to participation in the analysis and gave permission for his data to be published.

## 3. Results

### 3.1. Laboratory Examinations: Morphologic Stress Hormones 

Apart from slight deviations in the level of white blood cells, neutrophils, lymphocytes, and monocytes, there was a noticeable decrease in hemoglobin (Hb) and hematocrit levels one day after the run (Table 3 and Table 4).

### 3.2. Cardiopulmonary Exercise Stress Test (CPET) and Body Composition Assessment 

More information about Cardiopulmonary Exercise Stress Test (CPET) and Body Composition Assessment, please refer to Table 5, Table 6 and Table 7.

### 3.3. Echocardiography

One day after the 24-h UM, we observed an increase in left ventricle volume without a decrease in systolic performance, EF, or GLS assessment results. Indicators of diastolic function (E/e’ ratio, e’ wave) remained unchanged. An increase in the right ventricle size was noticed with a slight decrease in RV performance (s’RV) and an increase in RAA. All these changes returned to baseline after 10 days of recovery. Despite these differences, all the evaluated parameters remained within the normal ranges and were consistent with the physiology of physical training (Table 8).

### 3.4. Analysis of Mitochondrial Genome

In this study, the mtDNA sequence variation was investigated. The mitochondrial genome of the patient was covered by an average sequencing depth of 2850 ± 918 X. A total of 21 variants were identified with NGS and based on their full set, athlete’s mtDNA was assigned to HV0a1 haplogroup (Table 9).

Our subject carries haplogroup HV0a1, which is consistent with results of the Polish population published previously [43]. Additionally, the athlete bears 21 variants (single nucleotide or small deletions) located in control region (11/21), in genes coding 12S i 16S rRNA (3/21), as well as single mutations in ND2, COI, ATPase6, COIII, ND3, CYB and tRNA Thr (Table 9).

The m.9804G > A variant in *MT-CO3* gene has been associated with Leber hereditary optic neuropathy and multiple sclerosis; however, its pathogenic character was not verified by functional studies. Using pathogenicity prediction algorithms, the variant was assigned as likely benign. All except four variants, were homoplasmic or nearly homoplasmic and present in all or almost all copies of mtDNA. Four heteroplasmic variants were located in the m.303_315 region of mtDNA that covers polyC stretch and its length variations. Although three of them have been assigned to have low population frequency (below 0.5% frequency in the current set of GenBank sequences in MITOMAP), according to our experience, they were not rare. This discrepancy might be caused by technological limitations associated with different sample analysis or incorrect/different variant assignment in sequences deposited in GenBank. Besides those, there were only three relatively rare variants (of frequency below 0.5%), one in non-coding region of mtDNA (m.227A > G), one synonymous in the gene encoding ND3 subunit of respiratory chain complex I (m.10196C > T), and one nonsynonymous in the gene encoding CO3 subunit of complex IV (m.9804G > A). We did not observe any additional products on LR-PCR indicating the presence of large deletions of mitochondrial genome nor they were suggested in bioinformatic analysis of NGS data.

### 3.5. Psychological Examination

Below are presented the raw results of the respondent in particular variables, describing them with standard norms. The sten scores are standardized and correspond to the intensity of a given trait in relation to the general population. Stens 1st–3rd were low, 4th–7th average, and 8th–10th the highest.

#### 3.5.1. Personality Inventory (NEO-FFI)

In the five personality traits scales (Table 10), the respondent obtained an average result of neuroticism (5th sten); this implies normal level of anxiety, social anxiety, infrequent irritation, self-criticism, and emotional adaptation. On the extraversion scale, he obtained the 9th sten in relation to the male population of similar age, which is associated with a high level of the trait. An extroverted person is cordial, sociable, and open to social contacts. Openness to experience turned out to be a distinguishing feature of the respondent, in which he achieved the highest results (10th sten). A high score is equivalent to a positive evaluation of life experiences, cognitive curiosity, and openness to all novelties. Agreeableness of the competitor presented to a high degree (9th sten). This implies a positive attitude towards people, modesty, and avoiding conflicts. Demonstrated conscientiousness or the will to strive for achievements also manifested itself at a high level (9th sten). High degree conscientiousness was characterized by good organization, persistence, and individual motivation in those activities that were goal-oriented.

#### 3.5.2. The Sense of Coherence Questionnaire (SOC 29)

The respondent obtained the highest results for all three variables (Table 11). Sten 10th—the highest comprehensibility, was revealed in an ordered way of perceiving information. It was associated with a sense of predictability of situations and their better understanding. The second scale in which the respondent achieved a very high score, also the 10th sten was manageability. The construct was related to the belief in remedial resources. A high score indicated the sense of competence and the availability of external support, which helps a person to cope with all problems. The result of the meaningfulness scale was also a very high result (10th sten). A high intensity of the trait favored the tendency to give meaning to life events and the belief that it is worth putting effort and commitment to meet the requirements of life. This component of the sense of coherence is motivational and emotional.

#### 3.5.3. Life Orientation Test (LOT-R), the Hope for Success Questionnaire (KNS), the Generalized Self-Efficiency Scale (GSES)

Considering the scales of various tools that contain personality characteristics, the subject obtained high results in each of the scales (Table 12). In self-efficiency, 9th sten testifies to the belief in one’s own high abilities and competences, and in taking up difficult challenges. The test subject, in the ability to find solutions, obtained a result that was on the 9th sten. Such a high intensity of the feature was the disposition to pathway (ability to find solutions) in a problematic situation. In the agency variable (willpower), the ultra-runner also presented the result, placing himself on the 9th sten, which showed a very high sense of effectiveness in achieving his goals. The overall result including the above two components (hope for success) showed the highest result in relation to the population on the 10th sten. A person who obtained such a result is characterized by high self-confidence and a sense of favorable circumstances in the situation of undertaking actions. This was an extreme result, which may also indicate a propensity to take risks and too courageous behavior in unknown and uncertain conditions. The respondent reached the 8th sten on the optimism scale. This high result is related to the tendency to experience positive emotions and satisfaction with life.

#### 3.5.4. Scheler Value Scale (SVS)

Figure 1 presents important values by the respondent. Values guide actions and people’s behavior. The highest are the truth values, which the respondent revealed at the highest level (10th sten), obtaining a result close to the maximum number of points. This means following in life, among others: intelligence, wisdom, objectivity, open mind, broad mental horizon, and knowledge. High in the hierarchy of respondents’ values were moral values (9th sten), which have a numerical value similar to the truth value. Respecting moral values means commitment to goodness, honor, truthfulness, reliability, honesty, courtesy, and kindness. The athlete also considered vital values (8th sten) to be important in life; they include resistance to fatigue, physical strength, body elasticity, and the ability to endure cold and hunger. The respondent showed other results at an average level. The 7th sten was shown for aesthetic values, 6th sten for hedonistic values, and 5th sten for the sacred.

#### 3.5.5. Scale of Mental Toughness in Sport

Considering the results of the UM runner in each scale of mental resistance, the respondent obtained the maximum score in the variable self-confidence (Table 13). On the scale of effectiveness, the result turned out to be almost maximum at 94%. The subject achieved an average score on the emotional control scale, 62% of the possible maximum score. The implementation of tasks, which conditioned consistency in achieving the set goals, turned out to be scored the highest (100%).

#### 3.5.6. Perfectionism in Sports Questionnaire

Considering the results of the two scales of the perfectionism variable, the runner achieved positive perfectionism at the level of 100%, which indicates the maximum level of favorable aspirations (Table 14). On the negative perfectionism scale, the respondent obtained 36%, which means a relatively low result. These results determine an objective approach to one’s own abilities, as opposed to unrealistic, extreme requirements towards oneself, which turn out to be harmful to health.

#### 3.5.7. The Live Story Interview

The two modalities of communion and agency were indicators of significance in the interview.

Competitor coherently associated agency with close relationships. He was associated with people who showed similar values and sports interests. He had deep respect for the sports staff, including technical people (service technicians) and other players.

“*The most important are those closest to me—service technicians who take time off from work due to my start.*”

One of the first topics of the agency in the history of the respondent seemed to be active participation in middle childhood and early school age in intercity football matches (sports activity among adult cousins). In adolescence, the improvement in running resulted in the sports class and a more aesthetic and sporting silhouette were associated with a better sense of control, and thus predictability and safety. This was the cause of the “contamination sequence” in the event of lack of qualifications for a sports camp and the coach’s critical attention to the subject’s endurance capabilities.

The coach’s said that “*You are not going because you are too weak. It was the worst moment. I was training for two months during the summer holidays. Everyone was surprised. I remember that even my older colleagues were afraid of me*”.

The result of the trainer’s opinion was the “sequence of deliverance” in the form of individual training work, which (according to the respondent) led to very good running results in the period of high school and the respect shown by older colleagues. The key moments of victory were related in particular to achieving the set goal. The runner showed high determination and achievement motivation, while being able to defer gratification. In case of failure, he would be able to repeat the entire training procedure in the same sporting event in a year. He said that “*Even negative people are also great motivators. Some people just don’t like me because I’m winning*”.

In adulthood, the stories were often associated with numerous attempts to break the life record or take a specific place in the competition. He recalled the story when, after losing the competition, he had a conversation with the coach. “The coach was like a father. He never criticized me. It was an authority.”

Communion was expressed through relationships, few but deep friendships in the sphere of sports. “*You want someone to be proud of your start. You don’t want to let him down*”. In the period of early adulthood—during his studies, the respondent did not participate in any sports. He described these years as “a black moment in life”. “*I was a student who allowed himself to do anything. If my friend hadn’t called and didn’t say anything—we are reactivating the club—we are running; then, I would still eat fatty soups on Sundays. When I tried running 10 km for the first time, after the break, I had no strength to run*”.

A breakthrough event was the offer of a friend from the former running section to participate in weekly running meetings. The fact of being around people with sports led to regular participation in long-distance events, initially as technical support for athletes, and at the time of running long distances. “*You have to respect distance; you have to respect other runners and you should never focus only on winning*”.


*Communion can be expressed through the presence of an audience or the awareness that a larger group of people follow the achievements. “When there is great doping, you go faster.”*


Appropriate emotional regulation allowed him to separate the positive feelings directed at other runners from the willingness to compete and win a better place in the competition.


*“There are many reasons why I have achieved the results. One of the most important ones is humility towards the distance. I always start slower, giving myself time to tactically play my run later. The observation of opponents who, despite the fact that they have an advantage at this point, clearly signal their fatigue, gives me a very strong incentive to increase the pace.”*


The examined person showed high personal maturity and sports experience. To a large extent, he was the perpetrator (he trains himself, separates himself assertively from individuals who do not consider his needs and values, and isolates his superiors from his sports activities).


*“I believe that the most important element that guarantees my success is the possibility of good long-term preparation. The other elements are of much less importance. While running, I fear an injury that will eliminate me the most. My greatest organizational success is creating a competent team support who knows what they do during the competition and how to motivate me on the run. I am emotionally attached to him and I never want to disappoint him. We choose a goal together and implement it.”*


## 4. Discussion

### 4.1. Did the Obtained Results Identify “Champion Traits”?

The discovery of the “champion gene” and other physiological and psychological aspects that enable these athletes to achieve championship results is a desirable source of information not only for athletes, but also for their sponsors. The original goal of the pr esented research was to assess the impact of extremely high efforts on the heart and body of one of the world’s leading UM runners (in particular, in the 24-h run) [12]. None of the studies have shown damage to the heart from strenuous training and competition. Moreover, even the typical features described as “athlete’s heart” were not observed, in which one of the features is eccentric cardiac hypertrophy observed in echocardiography, resulting in increased stroke volume and ultimately increased VO_2_max. “Little heart” (Table 8), non-revelatory VO_2_max (Table 6), morphology and hematocrit in the lower normal values (Table 3) suggested a much wider aspect of being a champion than a simple “recalculation of results” from the diagnostic tests performed. The attempt to answer the question of what if “not the heart” influences the achievement of masterful results prompted us to extend the research to include: psychological tests, levels of stress hormones (three tests were all normal) and, above all, genetic tests. Interviews with the athlete have made it possible to “appreciate” the role of the support team, which although difficult to prove, can be essential for this athlete. The UM runner genetic study identified the mtDNA H haplogroup (HV0a1 subgroup, belonging to the HV cluster), which was observed more often among the top endurance athletes. However, it is quite common in the population of athletes not achieving such spectacular results. Psychological factors of the athlete indicated a high intensity of psychological properties tested. The conducted questionnaires analyzing personality traits indicated a high level of openness to experience and extraversion. The respondent also obtained an above-average result in the measurement of coherence, optimism, hope for success, and self-efficiency. In addition, the UM runner achieved the maximum result in positive perfectionism and a very high level of truth and moral values, which was confirmed by the interview. Runner showed a psychological predisposition to long-term effort, but an element that, being at a higher level, would help in even better affective regulation was emotional control (mental resistance component). The presented features were also not reserved only for champions and often occurred in people with incomparably smaller sports successes. It therefore seems that all these factors, as well as a number of others, contribute to the achievement of success in the 24-h run by our subject, and simple reductionism and an attempt to find an answer to the question that is also the aim of this study is still valid, although the obtained results suggest a “mosaic theory” (multifactorial, complex nature of the cause of achieved results or successes) to be the champion of the 24-h UM race.

### 4.2. To Be the Champion of the 24-h UM Race

There are many ingredients in the field of physiology, training tactics, psychology, environmental conditions, and other factors described in the following and previous sections which influence the final result in UM runs. They include age, specific aspects of anthropometry (low body fat), low BMI, and low limb and upper arm circumferences [44]. Personal best running times, extensive previous race experience, high running speed, and high running volume during training have importance [15]. When various anthropometric characteristics, such as skeletal muscle mass, body fat, and running training, were examined in multivariate analyses, only low body fat and fast running speed in training were correlated with fast race times [45]. In UM runners, weekly running distance and average running speed during training were negatively, and the sum of the skin folds positively correlated with race times [46]. Apart from aspects of anthropometry and training, age seems to have a major impact on UM performance. In 100-km UM runners, age, BMI, and body fat were positively and the weekly running distance negatively correlated to race time [47]. Low body fat and fast running speed during training are the most important predictors of a fast UM race time. Besides these variables, experience seems to be the most important variable for a successful performance in UM. UM runners need several years to reach their fastest running speed in a competition [48]. The personal marathon best time was an important predictor variable for mountain ultramarathoners [48]. In 24-h ultramarathoners, aspects such as anthropometric characteristics and training running volume showed no correlation with race performance, and the personal marathon best time showed the highest impact on UM race performance. To reach maximum distance in a 24-h UM, ultramarathoners should have a personal best marathon time of 3:20 h:min and have completed a continuous training run of at least 60 km before the race [49]. Only one of these conditions was met by our subject (best result in marathon: 2:42 h:min). However, anthropometric aspects such as low body fat or thin skin folds showed no relationship with race performance [49]. This fact is confirmed by the adipose tissue content in the body of our subject. The obtained 13.9% of body fat is an average value for marathon runners with less than 10% of body fat. At the same time, it should be remembered that in our research, the DEXA method was used, not bioelectric impedance analysis which lowers the results of the amount of adipose tissue, especially in athletes due to the greater amount of water in the body. Our data suggest that low body fat percentage is not a direct predictor of 24-h UM performance, and this assumption is supported by previous studies [50]. Waśkiewicz et al. showed a relatively high content of adipose tissue (13.9 ± 3.8%) in the group of UM runners (24-h UM) [50]. Other aspects were also investigated where cognitive functions should not be ignored. Faster ultra-marathoners seemed to focus on the relevant, unlike slower ultra-marathoners [51]. Another aspect is the pacing, or how ultra-marathoners distribute their energy across the race. In 24-h ultra-marathoners, the fastest runners start at lower relative intensities and display a more even pacing strategy than slower runners [52]. The runner we studied ran the first few kilometers in 24-h UM slowly (5:30/km), and the first 12 h were slower than the second. From the fourteenth hour of the run, he started to “fight” to maintain the pace and treated the last 2 h as an extended finish, running much faster than in the previous phase, always until the last second. Successful elite ultra-marathoners make almost no breaks during the race and often run in groups of runners of the same speed [53]. The only longer break that the runner took was 15 min and took place after 12 h of running (changing shoes, clothes, washing feet). Pacing has also been studied in few studies for age group runners. Taken together, pacing in an UM is crucial to finishing among the top [54]. Our athlete prefers to run 6–8 km behind top athletes for up to 12 h of running, possibly with another companion if he is running at the same preferred pace. During a UM, many problems occur, such as muscular spasms, overuse injuries, digestive problems, motivation problems, and sleep deprivation. Experienced ultra-marathoners have fewer medical problems such as muscle cramps and digestive problems.

For endurance sports performance, three main physiological factors appear to play the main role: maximal oxygen consumption (VO_2max_), LT, and efficiency (i.e., the oxygen cost to generate a given running speed or cycling power output) [5]. For an endurance athlete, physical talent is largely (but not exclusively) measured through the VO_2_ expressed as mL per kg of body weight per minute (mL/kg/min). Elite ultrarunners do not have VO_2_max values as high as elite marathon runners (around 77.4 mL/kg/min [55]), although they still have to be able to consume oxygen at or above a certain rate in order to compete at the top of the sport. In testing of elite ultrarunners, males can be successful with a VO_2_max of approximately 60 mL/kg/min and female athletes with a VO_2max_ of approximately 55 mL/kg/min [56]. This means that to win a big, competitive race, like the Western States 100, Leadville Trail 100, or Lake Sonoma 50, their VO_2_max values need to be around these values. In another examination of UM runners, the mean maximal aerobic power output (VO_2_max) of men was 72.5 mL·kg^−1^·min^−1^. These data suggest that success at the marathon and UM distances is crucially and (possibly) solely dependent on the development and utilization of a large VO_2_max [57]. At that time, less accurate oxygen uptake measurement tools were used than the current ones.

### 4.3. Potential Correlations between Mental and Physical Stress Markers and Performance in UMs

Little is known about the mutual correlation between the levels of certain hormones and neurotransmitters and the results obtained in sports, especially in UM runs. Perhaps one of the features of the “UM champion” enabling the achievement of outstanding results is the appropriate starting level and/or its variability during and after the run of substances referred to as markers of physical and mental stress. The group of mental and physical stress markers includes serotonin, cortisol, epinephrine, prolactin, testosterone, and luteinizing hormone (LH), among others, in our study. These substances are relatively easy to determine, possible to be estimated in any laboratory, and are characterized by relative stability, which was essential in the case of retrospective evaluation [58]. The possible influence of an exhaustive physical exercise on mental stress biomarkers (serotonin, tryptophan, and beta-endorphin) was examined by Agawa et al. in a 2-day UM race. An increase in serotonin, a drop in tryptophan, and an increase in β-endorphins were noticed. However, this study did not analyze the level and correlation of the marked markers (neurotransmitters) in individual phases of the UM played and the results obtained [59,60]. Ohta et al. conducted a study to clarify the effects of UM on the brain, examine the issue of central fatigue, verify the serotonin hypothesis of exercise-induced brain fatigue, and ascertain relationships between central fatigue and oxidative stress. There were no marked changes in serum serotonin levels in contrary to serum melatonin levels. The study suggests that running continuously for 24 h induces brain fatigue and that oxidative stress may be involved [60]. Pestell et al. examined individual hormonal responses to extreme physical stress in eight highly trained athletes. The length of a UM might influence the change in hormones [59]. In a run from Sydney to Melbourne over 1000 km, an increase of norepinephrine, epinephrine, dopamine, and adrenocorticotropic hormone was found. No changes were found for β-endorphin, growth hormone, prolactin, testosterone, cortisol, and cortisol-binding globulin. After the UM, catecholamines increased and growth hormone, prolactin, and cortisol increased. In conclusion, UM leads to a chronic physical stress leading to an adjustment of stress hormones with a continuous increase [59].

In other examinations, it is claimed that during UM, some characteristic changes in specific hormones are evident with the hypothalamic pituitary axis usually changing [61,62]. UM leads to an increase in cortisol, catecholamines, and growth hormone as well as a drop in testosterone where the decrease in testosterone is related to a decrease in libido [63]. According to Cleroux et al. increased free and total plasma dopamine levels have been noted in trained marathon runners correlating with better performance. The same levels of resting conjugated catecholamines are elevated in marathon runners and correlate with better performance [64]. Serum cortisol levels in runners have been found to be both elevated and within the normal reference range [65]. Villanueva et al. demonstrated elevated cortisol production in runners, presumably secondary to increased ACTH secretion [66]. Exercise certainly results in a rise in ACTH, and training augments this rise [67]. According to Tauler et al. (2014), shorter times to complete the ultra-endurance exercise were associated with higher increases in cortisol. However, no relationships were found between the time to complete the exercises and the changes in testosterone [68]. Male ultra-marathoners must be aware that increased levels of cortisol and suppressed levels of testosterone might become counterproductive [15]. Kupchak et al. assessed hormonal responses in men competing in the Western States Endurance Run (WSER): a 161-km trail run. Training for and completing the WSER produced a significant suppression in the hypothalamic-pituitary-testicular axis as seen by decreased levels of testosterone and LH. Additionally, running the WSER continued to influence endocrine function until 2 days after the race [69]. In a review article, Knechtle et al. points to the lower behaviors of hormones and neurotransmitters (not only stress markers) during UM. Increase was seen in cortisol, adrenocorticotropic hormone, noradrenaline, adrenaline, dopamine, growth hormone, prolactin, vasopressin, copeptin, aldosterone, atrial natriuretic peptide, N-terminal pro-brain natriuretic peptide, while decrease was seen in testosterone [15]. However, correlations between changes in the level of hormones and neurotransmitters described above, referred to as mental and physical stress markers, and their relationship with the results achieved in ultramarathon are still under investigation.

### 4.4. Morphology

The influence of morphology, and specifically of Hb, as an oxygen carrier is essential for aerobic exercise capacity. The close correlation between the level of Hb and VO_2_max is a documented relationship [70]. Blood tests (erythrocytes, Hb, hematocrit) are important elements of doping control [71]. In the case of our subject, the starting level of Hb was relatively low but normal, and its decrease in the test the day after the run below the normal level could not create an advantage over other UM runners.

### 4.5. A Search of Champion Genes: Mitochondrial Genome Variants

A fundamental question is the role of genetics in the attainment of world class status and truly elite athletic performance [9]. Several studies have reported that the key elements of the response to training in sedentary persons are widely variable and have a genetic component [8]. In this context, finding genetic markers that are strongly predictive of either success in endurance athletic performance or somehow preclude it is likely to be a challenging task because of the numerous cultural and environmental factors that contribute to success in sports, the physiological factors that interact as determinants of performance, and the heroic nature of the training required. Ideas about culture are highlighted by the observation that while East African runners currently dominated international competition, previously athletes from Australia and New Zealand, preceded by Eastern Europeans and even earlier, the Finns showed remarkable levels of success. This geographical diversity argues against a simple genetically based set of answers to the problem of elite performance in endurance competition.

As energy is required for multiple cellular processes, the mitochondria are essential in maintaining cell homeostasis, especially in high energy-demanding tissues such as skeletal muscle. It was shown previously that top-elite endurance athletes carry haplogroup HV0a1, which is consistent with results obtained by Maruszak and co-workers. They showed that haplogroup H (and HV cluster) is more frequent in top-elite endurance athlete group than power top-elite group [43]. Another interesting example relates to the gene encoding for the skeletal muscle isoform of AMP deaminase. There is a common mutation of this gene that may be associated with lower exercise capacity and “trainability” in untrained subjects [72]. While the frequency of the variant may be lower in elite endurance athletes, there are still a number of elite performers who carry it; hence, it does not appear to preclude the attainment of elite status and there is at least one example of an elite performer with essentially no AMP deaminase activity [73].

Sequence variations of the mitochondrial genome can affect the function of oxidative phosphorylation, especially nonsynonymous variants in protein coding genes. Such a role might be played by a relatively rare m.9804G > A variant found in the *MT-CO3* gene in mtDNA of the UM athlete. It was previously reported to be associated with diseases, however, its homoplasmic state and likely benign character as predicted by bioinformatic algorithms imply its rather mild impact on protein function. Even if so, it would be compelling to study the frequency of this variant among other athletes to see if it might be involved in athletic performance. However, variants (like m.16080A > G) previously identified in a group of elite endurance athletes from the Polish population have not been identified in the athlete’s mitochondrial genome [43].

The major non-coding region in mtDNA (control region, *MT-CR*) contains regulatory sequences associated with mtDNA replication and transcription, such as heavy strand replication origin (*MT-OHR*, m.110_441) and evolutionarily conserved sequence block (*MT-CSB2*, m.299_315). CSB2 is involved in the initiation of mtDNA replication. The transcription of the polyC tract within CSB2 (m.303_315) leads to the creation of a hybrid structure between RNA and DNA that causes its termination. The resulting structures (R-loops) provide free 3’ ends for initiation of DNA synthesis [74]. It was shown that a longer polyC tract correlates with a higher frequency of transcription termination. Therefore, it is suggested that the heterogeneity of the polyC tract length may be a mechanism modulating the formation of R-loop structures and, consequently, regulating the replication initiation process and mtDNA copy number [74,75]. A common transition T > C on position m.310 that is present in our runner causes formation of a continuous m.303_315 polyC stretch. It shows high structural stability and together with insertions expanding polyC tract, as found in the athlete, might be associated with efficient mtDNA replication, and consequently high energy production required during physical effort. The precise determination of the significance of the remaining variants requires more detail in silico analysis and comparison with the variants identified in athletes from various disciplines. These variants have not been reported in the context of genetic diseases and simple bioinformatic analysis does not indicate pathogenic. In addition, no large mtDNA rearrangements have been identified in our subject.

### 4.6. Fatigue, Pain, and Sleep Deprivation Resistance

UM races have become increasingly popular in the last few years in many countries. The ability to run long distances have played a role in human evolution, which makes ultralong-distance physiology important [76].

In measuring fatigue, both the perception of effort and the decline in force that occurs during sustained exercise should be addressed. This study aimed to propose a model that integrates the “neuromuscular” and “physiological” factors of fatigue (responsible for maximal force reduction) in UM running to explain the performance. Fatigue can be understood as a highly regulated strategy conserving cellular integrity, function and, indeed, survival. The flush model, dedicated to integrating the fatigue mechanisms in UM performance (and more generally to any type of endurance performance), using a holistic approach, is fully in agreement with this statement [77]. From the point of view of psychology, the attitude of the person who makes a sustained effort is important. Resilience manifested in coping with high physical stress and positive perfectionism that helps to achieve goals creates an important role in long-distance running [77]. According to Hoffman et al., in addition, faster runners experienced less pain in a 100-mile UM, in the sense that these ultra-marathoners had a better exercise-induced analgesia [78]. Our subject was characterized by high pain resistance. Pain in the right knee was experienced only after 12 h of running (approximately 130 km). The overload changes he experienced and which he struggled with while running are described in a separate article [18]. The increasing pain did not make him stop running, although it must have had an impact on the final result. Despite this, our subject finished the race with a victory [18]. Management of sleep deprivation seems to be central in long to very long UMs. Runners who adopted a sleep management strategy based on increased sleep time before the race completed the race faster [79]. During the 24-h run, athletes never sleep, but sleep resistance appears to play a role, although this is difficult to prove.

### 4.7. Motivational and Psychological Factors

A successful UM finish is a good life experience, while a non-finish is a big disappointment [80]. Motivational self-talk can be an effective tool to cope with exertion, as well as other stressors such as blister discomfort and adverse conditions [81]. This type of motivation was used by our subject.

The interview confirms and extends the collected information, which was included as the results of the conducted research with standardized questionnaires. The respondent showed high extrovertism and openness to experience, which is conducive to self-improvement and a positive attitude to new situations and interpersonal contacts (it does not matter in the context of sports performance, but in pursuing specific activities that meet the needs) [82]. It is similar with coherence, despite the lack of recorded correlation with sports results, the player with higher intensity of the variable has a greater sense of satisfaction and eudaimonic experiences [83].

In terms of professed values, in the interview, the competitor emphasized adherence to the prevailing general principles and care for social norms, which is confirmed by the high result of Scheler’s moral values (10th sten). He considered his sporting performances important for the country. The respondent also paid attention to the educational issues of sports, respect in relation to other (especially the elderly experience) runners. He expressed humility in describing his successes and failures. The key moments of victory were related, in particular, to the achievement of the set goal (as shown by the scale of implementation of mental resistance tasks. As reported by Cowden et al., high mental resilience promotes higher perfectionism (as revealed by the study), higher motivation, and setting optimal goals [84]. The UM runner was motivated by his training staff. In difficult situations, he did not want to disappoint these people that he cared most about. It is difficult to choose one property as determining a sports result. It is probably the combination of selected psychological features that contributes to such a strong pursuit of sports success.

### 4.8. Team Support

Successful finishers tackle the competition in small stages, paying attention to running speed, nutrition, hydration, and team support [80]. There is less information about the influence of team support taking care of the competitor during the run on the result obtained at the finish line. In many sports, such as long-distance swimming, trying to achieve your goal without team support is completely impossible [16]. According to the information from the competitor, the direct factors before the start that affect the final success in the 24-h run include various factors, among which the support he received from the support team is estimated at 60% (others are the preferred weather, season, home, and work situation), comfort of traveling to competitions, and several other factors. The team was a team of 4 consisting of people with whom the player cooperated for 4–10 years. They include a person responsible for preparing meals and drinks, controlling the time of the competitor and opponents, a person responsible for serving meals and changing clothes, and maintaining hygiene during the run. According to the surveyed competitor, even the best UM runner ends after 12 h if he does not receive very strong support from the team not only in terms of physiological needs (food, drink, clothing), but above all mental support, which is a specific form of motivation to continue the effort in superhuman loads. Perhaps it is the contribution of the support team that greatly contributes to such success for this athlete. Checking the contribution of such a team is more difficult than comparing the Vo2max achieved by individual players and should certainly be the reason for further research.

### 4.9. Mosaic Theory of Being a 24-h UM Race Champion—Definition?

More than 60 years ago, Dr. Irvine Page proposed the Mosaic Theory of hypertension, which states that many factors, including genetics, environment, adaptive, neural, mechanical, and hormonal perturbations interdigitate to raise blood pressure. In the past two decades, it has become clear that common molecular and cellular events in various organs underlie many features of the Mosaic Theory [85].

The authors of this article believe that many causes and mechanisms also contribute to the success of the competitor described here. None of the tested features or conditions (genetic, biochemical, anthropometric, environmental, psychological, cardiorespiratory system or hormonal system) is individually significantly different from the norm, or “reserved” for the tested competitor. If so, it can be assumed that each of the analyzed factors, regardless of or in combination with other factors, resulted in the ultimate possibility of achieving for many years and many times world-class championship results in the 24-h UM race. But still with a “?” mark.

### 4.10. Limitations

An important limitation was that the stress hormone tests were performed 24 h after the run. The dynamics of changes (the highest fluctuations during the effort and immediately after its completion) make them less useful as a determining factor for our subject’s success. On the other hand, such a term of research is justified because of easily measurable and repeatable characteristics of success in different periods from the start.

### 4.11. Perspectives

The performed tests should be repeated on a larger group of competitors achieving the best results at this distance. It is understood that the “master” is one, which makes finding a comparable large group for research difficult. However, it would be important to evaluate the characteristics of the leading players (e.g., the top five) in several events and compare them with the control group. Such an analysis would be a better documented attempt to answer the question on the repetitive features characterizing “the best UM runners”.

## 5. Conclusions

Our subject showed several features characteristic of champions in UM runs: genetic predisposition, mental traits, level of training, and resistance to pain. However, none are reserved exclusively for “champions.” Team support cannot be underestimated. The factors that guarantee the success of this elite 24-h UM runner go far beyond simple physiological and psychological explanations and remain a mystery in the search for individual elements of the probably “mosaic theory of being a champion.”

## Figures and Tables

**Figure 1 ijerph-18-02371-f001:**
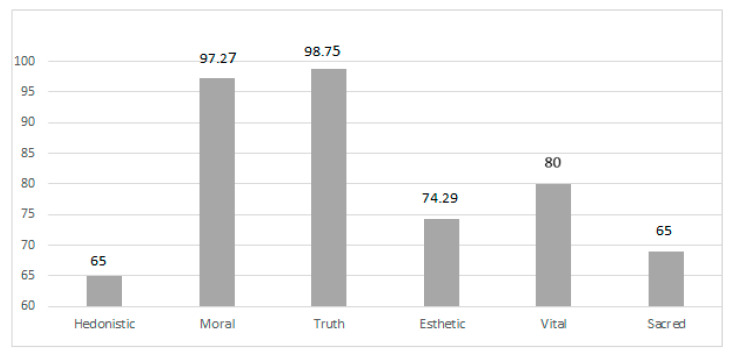
The hierarchy of values of the ultramarathon runner.

**Table 1 ijerph-18-02371-t001:** Athlete’s top 10 races.

Athlete’s Top 10 Races.				
Date of the Competition	Name of the Competition	Distance	Time	Open Place
9–10 June 2012	Polish Championships in 24-h run (POL)	235.542 km	24 h	III
8–9 September 2012	IAU 24-h WC/EC Katowice (POL)	249.809 km	24 h	X
19–20 October 2013	Polish Championships in 24-h run (POL)	251.113 km	24 h	I
26 April 2014	International Rudzki run 12-h (POL)	145.572 km	12 h	I
26–27 September 2014	Spartathlon (GRE)	245.3 km	25:49:05 h	III
30–31 May 2015	Ultrabalaton 221 km (HUN)	221 km	18:58:32 h	II
30 September–1 October 2016	Spartathlon (GRE)	246 km	23:02:23 h	I
8–9 April 2017	Polish Championships in 24-h run Łodz (POL)	256.246 km	24 h	I
26–27 May 2018	IAU 24-h EC Timisoara (ROU)	265.419 km	24 h	I
18 January 2019	48-h Athens Int. Ultramarathon Festival (GRE)	362.000 km	48 h	I

**Table 2 ijerph-18-02371-t002:** Date and type of medical tests.

Days before or after the Run	Before Run	1 Day after Run	10 Days after Run	Later
TTE	x	x	x	
Blood tests	x	x	x	
Cardiopulmonary exercise test	x			
Body composition	x			
Genetic test				x
Psychological tests				x

TTE, transthoracic echocardiography.

**Table 3 ijerph-18-02371-t003:** Summary results of blood tests.

Parameters	Units	Before the Run	1 Day after the Run	10 Days after the Run	Reference Values
Morphology					
White blood cells	10^9^/L	4	↑ 10.87	5.36	4.0–10.0
Neutrophils	10^9^/L	↓ 1.81	↑ 8.46	2.67	2.5–5.0
Lymphocytes	10^9^/L	↓ 1.22	↓ 1.18	1.74	1.5–3.5
Monocytes	10^9^/L	0.44	0.72	0.54	0.2–0.8
Eosinophils	10^9^/L	↑ 0.46	↑ 0.49	0.36	0.04–0.40
Basophils	10^9^/L	0.07	0.02	0.05	0.020–0.100
Red blood cells	10^12^/L	4.96	4.39	5.02	4.1–6.2
Hemoglobin	g/dL	15	↓ 13.4	15.2	14.0–18.0
Hematocrit	%	42.8	↓ 38.0	44.2	40.0–54.0
Mean corpuscular volume	fl	86.3	86.6	88	77.0–95.0
Mean corpuscular hemoglobin concentration	g/dL	35	35.3	34.4	32.0–36.0

**Table 4 ijerph-18-02371-t004:** Summary results of physical and mental stress markers.

Stress Marker	Before the Run	1 Day after the Run	10 Days after the Run	Reference Values
Serotonin (µg/L)	154.5	123.2	129.1	117.5–193.3
Cortisol (µg/dL)	15.57	12.52	14.61	5.27–22.5
Epinephrine (ng/L)	58	38	21	4.0–82.0
Prolactin (PRL) (ng/mL)	3.2	3	3.2	2.1–17.7
Testosterone (ng/dL)	745	398	476	241–827
Luteinizing hormone (mlU/mL)	4.25	2.12	1.68	1.50–9.30

**Table 5 ijerph-18-02371-t005:** Values of body mass and selected variables of body composition before competition.

Variable	
Body height (cm)	173
Body mass (kg)	63
BMI (kg/m^2^)	21.05
Bone mineral content (kg)	2.62
Fat mass (kg)	8.8
Lean mess (kg)	51.6
Body fat percentage (%)	13.9

**Table 6 ijerph-18-02371-t006:** Value of the maximal and lactate threshold workload and selected cardiorespiratory variables registered during Cardio-Pulmonary Exercise Test CPET before the competition.

Variable	
g_max_ at 14 km/h (%)	7.5
WR_max_ (W)	348
WR_max_ (W/kg)	5.5
VO_2max_ (L/min)	3.98
VO_2max_ (mL/kg/min)	63
g_LT_ at 14 km/h (%)	2.5
WR_LT_ (W)	277
WR_LT_ (W/kg)	4.4
VO_2LT_ (mL/kg/min)	49
VE_max_ (L/min)	158
RER_max4,1_	1.1
HR_max_ (bpm)	185

g_max_—maximal grade at 14 km/h during CPET, WR_max_—maximal workload during CPET, VO_2max_—maximal oxygen uptake, g_LT_ at 14 km/h—grade at lactate threshold, WR_LT_—workload at lactate threshold, VO_2LT_—oxygen uptake at lactate threshold, VE_max_—maximal ventilation, RER_max_—maximal respiratory ratio during CPET, HR_max_—maximal heart rate.

**Table 7 ijerph-18-02371-t007:** Changes in blood lactate concentration and values in selected variables of acid-base balance during CPET performed before the competition.

Variable	
LA_rest_ (mmol/L)	1.39
LA_max_ (mmol/L)	8.49
ΔLA (mmol/L)	7.1
pH_rest_	7.439
pH_max_	7.364
ΔpH	0.075
BE_rest_ (mmol/L)	−0.5
BE_max_ (mmol/L)	−7.0
ΔBE (mmol/L)	−6.5

LA_rest_—blood lactate concentration at rest, LA_max_—maximal blood lactate concentration during CPET, ΔLA—delta values of blood lactate concentration during CPET, pH_rest_—blood pH values at rest, pH_max_—blood pH values after CPET, ΔpH—delta values of blood pH changes during CPET, BE_rest_—base deficit at rest, BE_max_—base deficit after CPET, ΔBE—delta values of base deficit during CPET.

**Table 8 ijerph-18-02371-t008:** Heart systolic function in echocardiographic parameters.

Parameters	Units (Normal Values)	Before the Run	1 Day after the Run	10 Days after the Run
Left ventricle end-diastolic diameter volume	mL (106 ± 22)	109	126	113
Left ventricle end-systolic diameter volume	mL (41 ± 10)	33	35	33
Ejection fraction 2D (%) bi-plane	% (62 ± 5)	70	72	71
Global longitudinal strain	% (−20)	20.3	21.9	20.3
Interventricular septum diameter	mm (6–10)	9	10	10
Posterior wall diastolic diameter	mm (6–10)	9	9	9
Right ventricular end-diastolic diameter	mm (20–30)	31	34	29
S’ right ventricle	cm/s (14.1 ± 2.3)	16	14	17
Left atrium	mm (30–40)	33	36	34
Left atrial volume index	mL/m^2^ (16–34)	31.8	32.3	33.5
Right atrial area	cm^2^ (16 ± 5)	17.4	20.7	18.7

**Table 9 ijerph-18-02371-t009:** Mitochondrial variants and their heteroplasmy identified in the athlete by next generation sequencing (NGS).

mtDNA Variant	Heteroplasmy Level	*Locus*	Effect	GenBank Frequency	Comments
m.72T > C	100%	MT-CR	non-coding	1.77%	
m.195T > C	100%	MT-CR	non-coding	19.51%	
m.227A > G	98%	MT-CR	non-coding	0.33%	
m.263A > G	99%	MT-CR	non-coding	94.86%	
m.302_303insC	75%	MT-CR	non-coding	0.00% *	
m.302_303insCC	6%	MT-CR	non-coding	0.33%	Very rare
m.310T > C	8%	MT-CR	non-coding	40.43%	Potential hypertrophic cardiomyo-pathy (HCM) protective; negatively associated with obesity in Arab population
m.310_311insC	81%	MT-CR	non-coding	0.00% *	Potential hypertrophic cardiomyopathy (HCM) aggravating
m.514_515delCA	98%	MT-CR	non-coding	23.84%	Potential HCM aggravating
m.750A > G	100%	MT-RNR1	rRNA	98.27%	
m.1438A > G	100%	MT-RNR1	rRNA	94.82%	
m.2706A > G	97%	MT-RNR2	rRNA	78.94%	
m.4769A > G	99%	MT-ND2	p.Met100=	97.57%	
m.7028C > T	99%	MT-CO1	p.Ala375=	80.76%	
m.8860A > G	100%	MT-ATP6	p.Thr112Ala	98.47%	
m.9804G > A	100%	MT-CO3	p.Ala200Thr	0.29%	Associated in reports of literature with LHON and MS
m.10196C > T	99%	MT-ND3	p.Pro46=	0.03%	Very rare
m.15326A > G	100%	MT-CYB	p.Thr194Ala	98.65%	
m.15904C > T	99%	MT-TT	tRNA	1.59%	
m.16126T > C	100%	MT-CR	non-coding	11.29%	
m.16298T > C	100%	MT-CR	non-coding	6.68%	

LHON—Leber hereditary optic neuropathy, MS—multiple sclerosis. * The variant is not present in the current GenBank set of mtDNA sequences used by Mitomaster and MITOMAP database but is considered as frequent based on our earlier experience [25,26].

**Table 10 ijerph-18-02371-t010:** Characteristics of personality of the tested ultramarathon runner.

Neuroticism	Extraversion	Openness	Agreeableness	Conscientiousness
16	36	39	40	43

**Table 11 ijerph-18-02371-t011:** The sense of coherence of the ultramarathon runner.

Comprehensibility	Manageability	Meaningfulness
57	56	56

**Table 12 ijerph-18-02371-t012:** Personality variables of the ultramarathon runner.

Optimism	Self-Efficiency	Hope for Success	Agency (Willpower)	Pathway (Ability to Find Solutions)
20	37	61	30	31

**Table 13 ijerph-18-02371-t013:** Mental resistance variables.

Self-Confidence		Effectiveness		Emotional Control		Completing Task	
RESULT OBTAINED	Maximum Score	Result Obtained	Maximum Score	Result Obtained	Maximum Score	Result Obtained	Maximum Score
20	20	15	16	5	8	8	8

**Table 14 ijerph-18-02371-t014:** The results of perfectionism.

Positive Perfectionism		Negative Perfectionism	
Result Obtained	Maximum Score	Result Obtained	Maximum Score
65	65	31	85

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
