# Peer review of "To Be a Champion of the 24-h Ultramarathon Race. If Not the Heart ... Mosaic Theory?"

_ijerph, 2021, doi:10.3390/ijerph18052371_

Round 1

Reviewer 1 Report

Dear Authors

This is very interesting article which showed comprehensive evaluation before and after marathon using various methodology (echocardiography, comprehensive laboratory examinations including NGS, meticulous survey for mental and heath self reports). 

I have only one minor questions.

Blood sampling before marathon could be dangerous in relatively less-trained marathoners, what is the total amount of sampling?

And provide the rationale which justify to sample before the marathon, which could affect the result or even in health condition. 

I think that it doesn't any matter for the winner of marathoner, who is highly trained and naturally talented person.

But this study seems to be very interesting and the authors might want to expand this study for some more trained athletics. 

In that case, the recovery after blood sampling is minor concern. 

Author Response

Dear Reviewer:

Thank you very much for taking the time to review our article. We are very proud that the article was fully appreciated by your opinion. As you rightly pointed out, concern for the health of athletes and the result they want to achieve should require great caution in conducting any research. The volume of the blood sample taken for testing is approximately 10 ml. This amount has no effect on the athlete's health or the result achieved. It can also be safely collected from competitors of a much lower class before, during, and after the race.

Sincerely,

Robert Gajda and co-authors

Reviewer 2 Report

The evaluation of the publication is difficult. This is due to the inability to be clearly categorized.

The Introduction part presents the problem of research very inaccurately. On its basis, it is difficult to understand the justifications of research conducted on one athlete in the context of the set goal - "This comprehensive analysis aimed to identify the features enabling runners to achieve championship in 24-hour ultramarathon (UM) races".

Methodology

The research material is one athlete described in detail. However, this description does not constitute any starting point for further analysis. We get to know the athlete, his successes and sports career. It is difficult to understand why the link https://pzla.pl/aktualnosci/9409-10-mistrzstwa-polski-w-biegu-24-godzinnym) was not included in the list of literature. The cardiopulmonary exercise test (CPET) was performed on a treadmill. Table 6 shows the values in Watt and Watt / kg. This indicates that the methodology is inconsistent with the results or the use of conversion factors that have not been described in the research methodology. This is a serious methodological error. It may also indicate unreliability of the presented data. When describing the equipment, use the term "Kalifornia" (152) which is not found in English.

Results

The test results are presented inconsistently. Characteristics of morphologic stress hormones, body mass and selected variables of body composition, physiological and biochemical parameters have no description. All values are in the table. The values of the parameters Echocardiography and Analysis of Mitochondrial Genome are supplemented with a brief description. The values of psychological analysis are described very extensively (352-495). The psychological part takes up more than half of the text of the Results. It is not justified by the scope of the research and the number of collected data.

Discussion.

Extensive fragments of the discussion do not bear any reference to my own research. It is a descriptive review of the literature. The part concerning physiological and genetic issues is repeating the results of the research. It is difficult to call this part of the work a discussion. No comparison of own results with the results of other authors. One has the impression that the research was not preceded by an extensive analysis of the literature. Only the VO2max index was noted, other physiological indices were not discussed. The authors do not pay attention to the physiological indicators that are decisive for the result in UM (indicators related to the economy of running). The aspect of body composition is correctly presented in the discussion. The description of the results of psychological research is repeated again with reference to the works in which the same measurement tools were not used as the authors used. It is difficult to understand the willingness to compare research results obtained with different tools. The morphology is treated marginally in the discussion. There is no reliable study of this part of the research. The part relating to genetic testing is well worked out in the discussion. However, there are no reliable references to the literature on the subject of research. Later in the discussion, the authors return to psychological issues. This confirms the inconsistency of this part of the work. They also discuss a sociological problem that was not the subject of research.

Conclusion.

The conclusion comes down to the statement that the aim of the research was not achieved. This is a very original ending to such an extensive study.

It is not a full-scale research or experimental work. It is also not a review work, although it has many elements of such work. It is a research work based on a case study. The task set at the outset could not be solved at the level of one tested athlete. Ultarmarathon races require a specific mental structure (tests were not used to determine it), an efficient energy system (this has not been studied) and a locomotor system resistant to long-term operation (not shown) 597-599. Interestingly, it was on this element that another publication that had already appeared was based. This publication “Right Knee — The Weakest Point of the BestUltramarathon Runners of the World? A Case Study”, in contrast to the material presented for review, is defined as a case study. All the above-mentioned elements important for this type of research were known to the authors. It's hard to see why they haven't been researched. The work gives the impression of a mosaic composed of images, which, however, were not thought out to create one common composition. These images are random measurements from which the authors try to create a research paper.

Author Response

Dear Reviewer:  

Thank you very much for your effort and for the critical review of our manuscript.

 We agree with you that the evaluation of the publication is difficult . We realize  that our manuscript is much more extensive compared to a usual typical case study and that the variety of methods used may create a problem even for an experienced scientist and excellent reviewer.

We intend to use a case study as a material to broadly discuss or review the most important aspects of ultraendurance performance analyzed in our study (cardiopulmonary performance, psychology, genetics, markers of physical and mental stress, and so on). Certainly, we do not pretend to analyze and describe  in this case study all potential factors related to ultraendurance running and performance in 24-h ultramarathon. You are right that such factors, including indicators related to the economy of running, efficient energy system during run or locomotor system resistance to injuries or pain, were not analyzed; however, this was not the aim of our study. It seems that you do not appreciate the role of specific  psychological factors and team support as important factors limiting ultraendurance performance.

You are right that our paper is a research work based on a case study. Such methodological approach is fully justified and of scientific value even if “the problem could not be solved at the level of one tested athlete”. However, our paper may initiate the serious discussion about the problem and stimulate other research groups to perform similar comprehensive studies in large group of elite athletes.

We appreciate some of your critical remarks related to the methods of presenting and discussing our own results, but certainly we do not agree with your negative evaluation of the whole paper. The authors of the manuscript who represent leading centers and institutes dealing with sports sciences, sports medicine, cardiology,  physiology, biochemistry, and genetics have previously published successfully many hundreds of scientific papers in prestigious international journals. Many of our papers have got international recognition and hundreds of citations. In this context your last opinion that “These images are random measurements from which the authors try to create a research paper” is rather awkward.

We have not made any serious methodological error showing results of the CPET in Watt and in Watt/kg (texts highlighted in yellow):

Reviewer’s comment:
The cardiopulmonary exercise test (CPET) was performed on a treadmill. Table 6 shows the values in Watt and Watt / kg. This indicates that the methodology is inconsistent with the results or the use of conversion factors that have not been described in the research methodology. This is a serious methodological error. It may also indicate unreliability of the presented data

Authors reply:

We have corrected this information in the methodological section and we added the results in Table 6 as recommended

“To make the interpretation of the result easier to understand, the exercise intensity was also expressed in watts (W), as calculated by the MetaSoft software (Cortex, Germany) from the formula WR=(1,065+0,0511*g+9,322*10-4*g2) *v*w/4, were WR – workload in W, g – grade in %, v – speed in km/h, and w- weight in kg”

The formula was developed by Prof. Dr.-Ing. Christian Schulz of the Hochschule Mittweida, based on a formula of Magaria for the energy metabolism of runners. The Magaria formula was modified in such a way, that

  • The term representing the resting metabolism was excluded
    (because only the running performance is of interest)
  • The term representing the running energy metabolism was divided by 4 because the degree of efficiency is approximately 0.25.

In this way we get an estimation of the “running power”, which is equivalent to the power measured on bicycle ergometers.

We have used this formula in our previous studies:

 Czuba M, Bril G, PĹ‚oszczyca K, et al. Intermittent Hypoxic Training at Lactate Threshold Intensity Improves Aiming Performance in Well-Trained Biathletes with Little Change of Cardiovascular Variables. Biomed. Res. Int. 2019, 2019, 1287506. Doi:10.1155/2019/1287506

…………………………………………………

Thus, your comment suggesting unreliability of the presented data is not justified.

Thank you for comment related to the word “California “, we have corrected the error accordingly.

Sincerely,

Robert Gajda and co-authors

Reviewer 3 Report

This article makes a good contribution to the literature on what makes one an elite performer in ultrarunning. The methods, where a variety of measures are applied to one case, is very well done for the most part, and an approach that allows for much depth on a variety of related topics.

I’m not a physiologist or quantitative researcher, and have been assured the other reviewer can assess those aspects, so I’ll not say anything in regard to those. I will say, that in reading those methods, there were a number of times I wondered why the particular measures were being studied – I think more of a literature review could be presented overall (more on that in a minute).

The psychological methods were all well-verified tools, with appropriate adjustments made for the national context of Poland. When the data was being presented on those, the comments helped explain and situate the respondent very well.

I applaud the effort to include a more qualitative aspect to the study with the live story interview. Section 3.5.7, for the most part, presents useful data in an understandable way. I would suggest bringing more of the participants own words into that section. Typically in qualitative research, categories and themes are constructed, but then the participant’s own words are used to provide evidence for those. That’s currently lacking. Then, in the  discussion section on this (4.9), it’s not really a discussion, but a paragraph of quote. I’d suggest saving some of what is in 3.5.7 for the discussion part, and cut the quote in 4.9, but bring parts of it back as the evidence for the themes discussed earlier.  One other thought on this; at the end of the discussion of methods in 2.2.5, it’s a bit unclear what was done with the interview recording/transcript. It says (and there’s a typo in there that adds to the confusion, I think), “the recorded material (with the consent of the respondent) was divided into the suggested indicators’ communion) and agency.”  Does this mean that communion and agency were the themes uncovered? If so, then that needs to be explained better. Or are you using pre-existing categories? That would be more methodologically problematic and would need some justification. Clarify on that, please.

So this brings me back to an organizational issue. There’s really no literature review section in this paper. It goes from an explanation of the participant’s background, to study procedures, results and discussion. I realized later in reading the discussion that that’s why I kept asking the  “where did that come from” question that kept coming while reading the methods.  I’m suggesting that a lot of the discussion of literature that’s now in the discussion section, should be pulled out and reconstituted as a literature review before we get to study protocols. That would help with making clear why some of the measure were chosen.

The authors show a good familiarity with the literature, but much of it strikes me as being in the wrong place. Looking at the layout, there’s 88 citations, but fully 49 of them only come up for the first time in the discussion section. Within the discussion section, there’s wide swaths of material that’s talking about the extant literature, but not actually relating it back to this study.

Specifically:

- section 4.2. is a page and half of literature, with only a small aside about the participant on lines 571-572.  

- the entirety of section 4.3. is discussing literature and makes no referral back to the participant (or one so miniscule this reader totally missed it).

 - the first two paragraphs of 4.5 continue this trend. From there, the rest of 4.5 and all of 4.6, 4.7 and 4.8 do well to go back and forth between the literature and linking it to the study participant, which is how a discussion should be.

- 4.9 was mentioned above.

I would suggest that pulling some of that out to make a literature review would strengthen the paper in making it more readable overall, help set up the methods section and feed it through the quality data that’s there.

One last major thought – the whole idea of a “mosaic theory” warrants more discussion. At this point, it’s more a coining of a phrase than anything. Right now, it’s mentioned in the title, abstract, the end of the first paragraph (line 532), and the last line of the paper. It’s a neat idea, and could use least a couple of paragraphs to explain what the authors mean by it, especially as its something that could be easily citable from this. The authors should define and explain it, so others don’t just run off in various directions with it.  

Minor things: there’s a couple of lines where it was a bit unclear, with either a word missing or the phrasing being hard to discern, and it seemed like key points were being made.

- line 68-69 – “professionalism of the team”?

- lines 266-267 mentioned above

- line 499 – “the champion gene” – something lost in translation?

- line 535 – “to be the champion of the 24 hour race”?

 - lines 600-601, lines 752-753 – probably needs a reference for those statements of fact.

Author Response

Dear Reviewer:

  Thank you very much for your effort and very positive evaluation of our paper. We know that our manuscript is much more extensive compared to a usual typical case study, and that the  variety of methods used might have created a problem. We very much appreciate your positive opinion about the variety of methods used, good familiarity with the literature (88 citations; after correction, the number of references is now 91), and especially detailed comments related to the psychological part of the study. 

In fact, our manuscript based on a profound analysis of several selected features related to the ultraendurance  performance in only one elite champion runner, is a novelty approach. Because of the variety of methods used to prepare a comprehensive and profound study as well as limited volume of the whole publication, we have to compromise some parts of the manuscript.

We intend to use a case study as a material to broadly discuss or review the most important aspects of ultraendurance performance analyzed in our study (cardiopulmonary performance, psychology, genetics, markers of physical and mental stress, and so on). Certainly, we do not pretend to analyze and describe  in this case study all potential factors related to ultraendurance running and performance in 24-h ultramarathon

Because of the variety of the methods used in our study and in order to guarantee quality results, as much as possible, of the analyzed features, we invited a relatively large group of experts from leading Polish (and Swiss) Institutes and centers, representing various professions and disciplines (i.e. sports medicine, sports sciences, cardiology, physiology, psychology, genetics, biochemistry, etc.) Thus, despite their competence and experience, finalizing the manuscript was not an easy task, and we agree with your specific comments suggesting necessary editorial changes in order to make the whole publication more readable and if possible more clear in some parts .

According to your suggestion, we have moved some parts of the Discussion to the Introduction.

Specific changes made in the revised version of the manuscript, according to almost all of your suggestions, are shown in blue font.

Among them, an additional subsection on the "mosaic theory": 4.9. Mosaic theory of being a 24-hour UM race champion — definition?

We also significantly changed the title of the article from the original one:

‘’To be a champion of the 24-hour ultramarathon race. If not the heart ...? Mosaic theory”

Revised title:

‘’To be a champion of the 24-hour ultramarathon race. If not the heart ... Mosaic theory?”

We believe that this way of presenting the numerous changes in the article will be the most readable for you. We attach version: manuscript with colors, for easier tracking of changes.

Despite some editorial imperfection of our manuscript, we hope that you accept the paper in the present revised form. We are sure that such kind of study deserves presentation in prestigious international journals and could inspire other research groups to conduct similar studies involving more numerous cohorts of elite athletes. We intend to continue our research in line with this, although it is very difficult  to collect real-life data during competitions and to incline World Champions and winners of most prestigious ultramarathons or other ultraendurance sports events to take part in long-lasting and time-consuming investigations.

Last but not the least, there are few elite athletes with many prestigious champion titles who are already in very competitive environment to share their biomedical and psychological data with others, including competitors.

Sincerely,

Robert Gajda and co-authors

Round 2

Reviewer 2 Report

The responses for review received are not unsatisfactory. The content contained is highly original and different from that used in the world of science. By stating:We realize that our manuscript is much more extensive compared to a usual typical case study and that the variety of methods used may create a problem even for an experienced scientist and excellent reviewer”  the authors suggest to the reviewer that their work is outstanding. This makes it very difficult to review. Unfortunately, the authors did not understand the reviewer's intention. The review noted that the work is inconsistent and it compares the data from the literature with its own observations in any way. This work is not a logically designed case study. It is not a review. It is a difficult hybrid to classify to be a work based on mosaic theory. The mosaic theory refers to a method of analysis used information about a corporation. The mosaic theory involves collecting public, non-public, and non-material information. Based on this information, it evaluates the value of the object it analyzes. Do the authors judge their respondents on the basis of the collected materials - no. Do they create the assessment of all ultramarathon runners - no. In the reviewer's opinion, the term the mosaic theory was overused Do we get an answer to the question "What are the key factors of elite 160 performance in 24-hour UM run?" Of course not, the work does not systematize the results in the direction of answering this question. It is still difficult to understand the meaning of converting running speed into power. The formula was given, but the reason for this conversion was not given. This would be understandable in triathlonist studies. In this case, it is difficult to see the sense. Citing own studies (self-citations) is not accurate

Extensive argumentation of the authors of the paper : “We appreciate some of your critical remarks related to the methods of presenting and discussing our own results, but certainly we do not agree with your negative evaluation of the whole paper. The authors of the manuscript who represent leading centers and institutes dealing with sports sciences, sports medicine, cardiology, physiology, biochemistry, and genetics have previously published successfully many hundreds of scientific papers in prestigious international journals. Many of our papers have got international recognition and hundreds of citations. In this context your last opinion that “These images are random measurements from which the authors try to create a research paper” is rather awkward.”  can be summarized in one task. We are such an outstanding group that we do not make mistakes and our publications do not require reviews. It can also be added that there are no reviewers who can evaluate such complex and perfect studies. However, as a reviewer, let me disagree with the authors. At the same time, I would like to inform you that I do not evaluate the authors' other publications..

Author Response

Dear Mr. Reviewer,

 Thank you for your thorough analysis and pertinent comments on the definition of "mosaic theory". The definition that you have given is obviously accurate but it concerns  the activities within the corporation (You: The mosaic theory refers to a method of analysis used information about a corporation. The mosaic theory involves collecting public, non-public, and non-material information. Based on this information, it evaluates the value of the object it analyzes). The term "mosaic theory" is not, however, reserved for this sphere, because this is the definition used by Dr. Irvine Page “… more than 60 years ago, Dr. Irvine Page proposed the Mosaic Theory of hypertension, which states that many factors, including genetics, environment, adaptive, neural, mechanical, and hormonal perturbations interdigitate to raise blood pressure. In the past two decades, it has become clear that common molecular and cellular events in various organs underlie many features of the Mosaic Theory (Frohlich, E.D.; Dustan, H.P.; Bumpus, F.M.; Irvine H. Page: 1901-1991. The celebration of a leader. Hypertension. 1991; 18, 443-445; DOI:10.1161/01.hyp.18.4.443.) There are also other definitions for "mosaic theories" concerning other areas of life (“The mosaic theory in FOIA law(1972-2001) — The mosaic theory is, essentially, a theory of informational synergy. It describes a process through which adversaries collect, combine, and compile items of information, some or even all of which are harmless in their own right. And it suggests an outcome whereby this process, in a feat of analytic alchemy, converts the harmless information into something useful…” — https://www.yalelawjournal.org/pdf/358_fto38tb4.pdf.). Accepting your suggestion that the presented case cannot be clearly named as "mosaic theory", we changed the title of the work from the original one:

“To be a champion of the 24-hour ultramarathon race. If not the heart ...? Mosaic theory”

to:

“To be a champion of the 24-hour ultramarathon race. If not the heart ... Mosaic theory?”

To address your critical assessment of the term "mosaic theory", we have provided OUR own definition of “mosaic theory”, describing it in an additional subsection (4.9. Mosaic theory of being a 24-hour UM race champion — definition?)

4.9. Mosaic theory of being a 24-hour UM race champion — definition?

More than 60 years ago, Dr. Irvine Page proposed the Mosaic Theory of hypertension, which states that many factors, including genetics, environment, adaptive, neural, mechanical, and hormonal perturbations interdigitate to raise the blood pressure. In the past two decades, it has become clear that common molecular and cellular events in various organs underlie many features of the Mosaic Theory [91].

The authors of this article believe that many causes and mechanisms also contribute to the success of the competitor described here. None of the tested features or conditions (genetic, biochemical, anthropometric, environmental, psychological, cardiorespiratory system, or hormonal system) is individually significantly different from the norm, or "reserved" for the tested competitor. If so, it can be assumed that each of the analyzed factors, regardless of or in combination with other factors, resulted in the ultimate possibility of achieving for many years and many times world-class championship results in the 24-hour UM race. However, it still with a “?” mark.

The question marks at the beginning and end of the section, as well as in the title of the article, pose a question and do not affirmatively state the reason for the results achieved by the tested competitor. In the subsection "Perspectives" we indicated the need for further research to determine the reasons for other runners achieving outstanding results in ultramarathon runs.

 4.11. Perspectives

The performed tests should be repeated on a larger group of competitors achieving the best results at this distance. It is understood that the “master” is one, which makes finding a comparable large group for research difficult. However, it would be important to evaluate the characteristics of the leading players (e.g., the top five) in several events and compare them with the control group. Such an analysis would be a better documented attempt to answer the question on the repetitive features characterizing "the best UM runners."

I hope that my responses to the points that you have raised regarding the CPET test will satisfy you.

Reviewer:  It is still difficult to understand the meaning of converting running speed into power. The formula was given, but the reason for this conversion was not given. This would be understandable in triathlonist studies. In this case, it is difficult to see the sense. Citing own studies (self-citations) is not accurate

Answer:

This part has been changed in the article:

Given that the inclination on the treadmill was increased during the CPET test, and to make the interpretation of the result easier to understand for coaches, the intensity was expressed as velocity (v) with grade (g), and power (WR). Power was calculated using the MetaSoft software (Cortex, Germany) with the following formula: WR=(1.065+0.0511*g+9.322*10-4*g2) *v*w/4, where WR is the workload in W, g is grade in %, v is velocity in km/h, and w is weight in kg.

Reply:

To address the Reviewer's comment, information on treadmill velocity and grade has been added in Table 6. The presentation of the results also in the form of workload expressed in Watts is necessary because the treadmill grade changes during the exercise test. This provides an opportunity for both researchers and coaches to refer to the aerobic power measured in the tests and to the energy expenditure. Measuring power generated during running to monitor exercise intensity has become increasingly common in recent years.

References:

Jaén-Carrillo D, Roche-Seruendo LE, Cartón-Llorente A, Ramírez-Campillo R, García-Pinillos F. Mechanical Power in Endurance Running: A Scoping Review on Sensors for Power Output Estimation during Running. Sensors (Basel). 2020 Nov 13;20(22):6482. doi: 10.3390/s20226482. PMID: 33202809; PMCID: PMC7696724.

Cartón-Llorente A, García-Pinillos F, Royo-Borruel J, Rubio-Peirotén A, Jaén-Carrillo D, Roche-Seruendo LE. Estimating Functional Threshold Power in Endurance Running from Shorter Time Trials Using a 6-Axis Inertial Measurement Sensor. Sensors (Basel). 2021;21(2):582. Published 2021 Jan 15. doi:10.3390/s21020582

Jaén-Carrillo D, Roche-Seruendo LE, Cartón-Llorente A, Ramírez-Campillo R, García-Pinillos F. Mechanical Power in Endurance Running: A Scoping Review on Sensors for Power Output Estimation during Running. Sensors (Basel). 2020;20(22):6482. Published 2020 Nov 13. doi:10.3390/s20226482

Cerezuela-Espejo V, Hernández-Belmonte A, Courel-Ibáñez J, Conesa-Ros E, Mora-Rodríguez R, Pallarés JG. Are we ready to measure running power? Repeatability and concurrent validity of five commercial technologies. Eur J Sport Sci. 2020 Apr 26:1-10. doi: 10.1080/17461391.2020.1748117. Epub ahead of print. PMID: 32212955.

Imbach F, Candau R, Chailan R, Perrey S. Validity of the Stryd Power Meter in Measuring Running Parameters at Submaximal Speeds. Sports (Basel). 2020;8(7):103. Published 2020 Jul 20. doi:10.3390/sports8070103

Arampatzis A, Knicker A, Metzler V, Brüggemann GP. Mechanical power in running: a comparison of different approaches. J Biomech. 2000 Apr;33(4):457-63. doi: 10.1016/s0021-9290(99)00187-6. PMID: 10768394.

Other corrections made on your suggestions:

Before correction:

This comprehensive analysis aimed to identify the features enabling runners to achieve championship in 24-hour ultramarathon (UM) races".

After correction:

"This comprehensive analysis aimed to identify the features enabling a runner to achieve championship in 24-hour ultramarathon (UM) races".

Your remark:

. It is difficult to understand why the link https://pzla.pl/aktualnosci/9409-10-mistrzstwa-polski-w-biegu-24-godzinnym) was not included in the list of literature.

 Correction:

I corrected the text and removed the redundant link ( https://pzla.pl/aktualnosci/9409-10-mistrzstwa-polski-w-biegu-24-godzinnym ).

 Your remark:

The authors do not pay attention to the physiological indicators that are decisive for the result in UM (indicators related to the economy of running).

The answer:

These indicators have not been studied (economy of running) and we have nothing to say in this regard. Independently, we did the research and analyzed the results only approved by the athlete. 

Your remark:

The morphology is treated marginally in the discussion.

Answer:

The morphology results are in the lower limit of the normal range and cannot have a positive effect on the race result.

We have significantly changed the Introduction as well the presentation of the results of the psychological tests (in the previous manuscript: after the first review [blue font color]).

The minor errors that you have pointed out in the first review have been corrected. However, it is impossible to correct the article title this time, as the other two reviewers have fully accepted the article title after the making the necessary corrections or after clarifying doubts. A profound reorganization of the article from its current state would certainly not satisfy the other reviewers and all co-authors.

I would also like to apologize to you for if my responses on the your first review sounded slightly too emotional.Of course, you were the one who evaluated this particular article; thus, your evaluation of our work cannot be criticized by the people being evaluated, i.e. the authors.Moreover, I am not an "outstanding author", but some of my co-authors who contributed to this paper are.My response to your comments was my personal reaction as a person who also felt responsible for the manuscript and who represents the good name of all the co-authors of this article and their indisputable position in the world of science.Please accept my apologies again.

I am hoping hope that the above explanations and corrections (yellow background) made to the article satisfy you to a large extent. Considering my responses and despite the shortcomings and imperfections of the work in your opinion, please consider accepting the work in this version. We believe that our manuscript is an important source of information about the supposed reasons for achieving outstanding results in the ultramarathon race by the described athlete and contributes to the search for an answer to the question “What factors contribute to achieving outstanding results in ultramarathon by others champions of this distance?”

Sincerely,

Robert Gajda and co-authors
